Drug delivery systems for ovarian cancer treatment: a systematic review and meta-analysis of animal studies

Raavé René 1
de Vries Rob B.M. 2
Massuger Leon F. 3
van Kuppevelt Toin H. 1
Daamen Willeke F. 1 Willeke.Daamen@radboudumc.nl
1 Department of Biochemistry, Radboud university medical center , Nijmegen , The Netherlands
2 Systematic Review Centre for Laboratory Animal Experimentation, Central Animal Facility, Radboud university medical center , Nijmegen , The Netherlands
3 Department of Obstetrics and Gynaecology, Radboud university medical center , Nijmegen , The Netherlands
Erez Offer
Electronic publication date: 2015 Dec 10
Publication date: 2015
Volume: 3
Electronic Location ID: e1489
Received 2015 Aug 14; Accepted 2015 Nov 19
Copyright: © 2015 Raavé et al.
Copyright year: 2015
Copyright holder: Raavé et al.
License: This is an open access article distributed under the terms of the Creative Commons Attribution License, which permits unrestricted use, distribution, reproduction and adaptation in any medium and for any purpose provided that it is properly attributed. For attribution, the original author(s), title, publication source (PeerJ) and either DOI or URL of the article must be cited.
License URL: https://creativecommons.org/licenses/by/4.0/

Keywords: Animal studies, Drug delivery systems, Ovarian cancer, Systematic review, Meta-analysis

Funding: Radboud university medical center The Netherlands Organisation for Health Research and Development 104024065 This project was funded by the Radboud university medical center. Rob de Vries received funding from The Netherlands Organisation for Health Research and Development (ZonMw; grant nr. 104024065). The funders had no role in study design, data collection and analysis, decision to publish, or preparation of the manuscript.

==============================
Current ovarian cancer treatment involves chemotherapy that has serious limitations, such as rapid clearance, unfavorable biodistribution and severe side effects. To overcome these limitations, drug delivery systems (DDS) have been developed to encapsulate chemotherapeutics for delivery to tumor cells. However, no systematic assessment of the efficacy of chemotherapy by DDS compared to free chemotherapy (not in a DDS) has been performed for animal studies. Here, we assess the efficacy of chemotherapy in DDS on survival and tumor growth inhibition in animal studies. We searched PubMed and EMBASE (via OvidSP) to systematically identify studies evaluating chemotherapeutics encapsulated in DDS for ovarian cancer treatment in animal studies. Studies were assessed for quality and risk of bias. Study characteristics were collected and outcome data (survival/hazard ratio or tumor growth inhibition) were extracted and used for meta-analyses. Meta-analysis was performed to identify and explore which characteristics of DDS influenced treatment efficacy. A total of 44 studies were included after thorough literature screening (2,735 studies found after initial search). The risk of bias was difficult to assess, mainly because of incomplete reporting. A total of 17 studies (377 animals) and 16 studies (259 animals) could be included in the meta-analysis for survival and tumor growth inhibition, respectively. In the majority of the included studies chemotherapeutics entrapped in a DDS significantly improved efficacy over free chemotherapeutics regarding both survival and tumor growth inhibition. Subgroup analyses, however, revealed that cisplatin entrapped in a DDS did not result in additional tumor growth inhibition compared to free cisplatin, although it did result in improved survival. Micelles did not show a significant tumor growth inhibition compared to free chemotherapeutics, which indicates that micelles may not be a suitable DDS for ovarian cancer treatment. Other subgroup analyses, such as targeted versus non-targeted DDS or IV versus IP administration route, did not identify specific characteristics of DDS that affected treatment efficacy. This systematic review shows the potential, but also the limitations of chemotherapy by drug delivery systems for ovarian cancer treatment. For future animal research, we emphasize that data need to be reported with ample attention to detailed reporting.

Introduction

Ovarian cancer is the most lethal of all gynecological cancers. It is estimated that approximately 65,500 women were diagnosed with ovarian cancer and that about 42,700 women deceased due to ovarian cancer in Europe in 2012 (Ferlay et al., 2013). Conventional therapy includes neoadjuvant chemotherapy with subsequent surgical interval debulking and subsequent chemotherapy or primary surgical debulking with adjuvant chemotherapy (Vergote et al., 2011; Vergote et al., 2010). Although systemic intravenous administration of chemotherapeutics results in elimination of cancer cells, it is associated with serious shortcomings. Chemotherapeutics have a short half-life, are toxic to healthy cells and show an unfavorable biodistribution resulting in undesired side effects such as bone-marrow suppression, neuropathy, cardiotoxiticy, hair loss and nausea (Bergkvist & Wengstrom, 2006; Chon et al., 2012; Love et al., 1989; Massey, Kim & Abdi, 2014; Monsuez et al., 2010; Truong et al., 2014). Moreover, next to systemic intravenous (IV) administered chemotherapy, local intraperitoneal (IP) in combination with IV administration is applied as well and was found to increase survival time in ovarian cancer patients (Armstrong et al., 2006; Barlin et al., 2012; Jaaback, Johnson & Lawrie, 2011), but these patients had more side effects. Drug delivery systems (DDS) may overcome the current disadvantages of chemotherapeutics. By encapsulating toxic chemotherapeutics, DDS are designed to increase concentrations of chemotherapeutics at the tumor site, which could eventually result in higher treatment efficacy, while simultaneously reducing exposure of chemotherapeutics to healthy cells, resulting in a therapy with reduced side effects.

To date, abundant research has been performed on DDS, which has resulted in many kinds of DDS, such as liposomes, micelles or ‘nanoparticles’ (Lammers et al., 2012; Tomasina et al., 2013), with different characteristics for treatment of various types of cancer, including ovarian cancer. Several factors may affect the efficacy of DDS. For instance, size can be of importance as for long blood-circulation times and optimal tumor penetration an optimal size range of DDS is estimated to be in the sub-100 nm, but not smaller than 6 nm to prevent unwanted removal (Perrault et al., 2009). Another parameter that is often varied among DDS is PEGylation, which is intended to prevent unwanted uptake by the liver and spleen by coating the surface of DDS with poly(ethylene)glycol (PEG) resulting in increased blood-circulation times (Ernsting et al., 2013; Perrault et al., 2009). With increasing circulation time, increased accumulation of DDS can be found at the tumor site. By the enhanced permeability and retention (EPR) effect of the tumor cell aggregates, due to leaky blood vessels, DDS accumulate in the tumor area and release their content, so-called passive targeting (Iyer et al., 2006). On the other hand, a more active way of targeting can be achieved by conjugating anti-tumor antibodies or specific receptor ligands to the wall of capsules to target tumor cells specifically (Danhier, Feron & Preat, 2010). The passive and active targeting strategies mainly apply to intravenously (IV) administered DDS. However, as IP administered chemotherapy in combination with IV administered chemotherapy is being clinically applied, DDS are also being administered IP instead of IV in ovarian cancer (Gunji et al., 2013; Ye et al., 2013; Zhang et al., 2013), introducing another variable in DDS that can affect the efficacy of DDS therapies. Furthermore, the DDS preparation material can be varied from metals to polymers to proteins, which influences properties such as biodegradability, immunogenicity and toxicity, but also drug release characteristics or cellular uptake of DDS. Various chemotherapeutics are entrapped in DDS for ovarian cancer treatment, such as cisplatin, paclitaxel or doxorubicin, affecting the outcome of DDS treatment as well. All in all, preclinical studies showed that many parameters can be varied in DDS. It is still unclear, however, which variant is most effective.

The majority of DDS are evaluated in vitro before being tested in animal models using different cancer cell lines. In vivo evaluation has shown a wide range of therapeutic efficacies, with different treatment regimes and several time periods. Several reviews describe possible improved efficacies that chemotherapy by DDS may have in animal models for cancers such as breast cancer (Yezhelyev et al., 2006), lung cancer (Loira-Pastoriza, Todoroff & Vanbever, 2014), melanoma (Cheng et al., 2014), brain cancer (Chen et al., 2013), colorectal cancer (De Smet et al., 2013) and ovarian cancer (Tomasina et al., 2013). A recent literature overview by Tomasina et al. (2013) showed a number of DDS that have been studied for ovarian cancer treatment. However, no systematic assessment of the efficacy of DDS in experimental ovarian cancer, or other cancer types, and the effects of the different characteristics of these DDS on treatment outcome has been reported. Therefore, we have conducted a systematic review of animal studies in order to gain insight into the effectiveness of the many types of DDS tested for ovarian cancer treatment.

In clinical studies, systematic reviews are common practice and they are also gaining popularity in preclinical (animal) studies. Compared to narrative reviews, systematic reviews are more structured and more thorough, resulting in a more comprehensive and transparent overview. Systematic reviews are therefore an ideal method for gaining a better understanding of the role DDS play in ovarian cancer therapy. Furthermore, such review may give new insights into the most effective capsule characteristics, how to improve the use and design of animal models, and eventually clinical trials. Moreover, meta-analysis can be used as an additional tool in systematic reviews of animal studies. While in meta-analyses of clinical data the primary goal is mostly to obtain a precise estimate of the overall effect of a certain intervention, in meta-analyses of animal studies the exact overall effect size may not be that informative (because of the often large heterogeneity between animal studies) and therefore the goal is of explorative nature to identify factors that affect the main outcome (Hooijmans et al., 2014a).

In this article, we report the results of the first systematic review of DDS evaluated in ovarian cancer animal models. In a comprehensive literature screening, we included all animal studies that used chemotherapeutics encapsulated in a DDS and evaluated their therapeutic efficiency in an orthotopic ovarian cancer animal model. A complete overview of the available literature including an assessment of the risk of bias of the individual studies is included. Where possible, meta-analyses were performed to study the extent to the efficacy of DDS depend on the different subgroup characteristics (type of drug delivery system, targeted vs. non-targeted DDS, IP vs. IV administration, type of xenografted cell line and type of chemotherapeutic in DDS).

Methods

Search strategy, inclusion and exclusion criteria

To include as many animal studies as possible on drug delivery systems for ovarian cancer treatment, a comprehensive search strategy for PubMed and EMBASE (via OvidSP) was developed. The search strategy consisted of three specific search components addressing: (1) drug delivery systems; (2) ovarian cancer; and (3) animal studies. The search strategy included thesaurus terms and keywords on the subject of drug delivery systems and ovarian cancer (see supplemental methods for complete search strings). To include all animal studies, previously developed PubMed and EMBASE search filters were used (De Vries et al., 2014; Hooijmans et al., 2010). No language restrictions were applied.

After the search strategy had been executed in PubMed and EMBASE (search up until September 1th 2014), duplicates were manually removed and the resulting studies were screened by title and abstract and classified as included, more information required or excluded, according to predefined exclusion criteria to exclude studies that did not comply with our research question (see supplemental methods for criteria). Included and more information required classified studies were subjected to a full text screening using additional exclusion criteria described in the supplemental methods. Screenings were performed independently by two reviewers (RR and WD) using Early Systematic Review Software 2.0 (EROS, Institute of Clinical Effectiveness and Health Policy, Buenos Aires, Argentina). Differences in classification between reviewers were discussed until consensus was reached. Studies in a language other than English (e.g., Japanese and Chinese) were screened by title and abstract by native speakers for that specific language. If a non-English study was included in the systematic review, it was professionally translated by “Radboud in’to Languages” (Radboud University, the Netherlands).

Study characteristics

Journal and author information from all included studies was registered. Drug delivery characteristics (e.g., material, size, etc.), animal model information (e.g., species, cell lines, etc.) and treatment and outcome characteristics (e.g., dose, regime, tumor size evaluation, etc.) were extracted. Conference abstracts and studies without data comparing free drug vs. encapsulated drug were not included in the meta-analyses. One study (Ueno, 1988) was not included in the meta-analysis as we were not able to identify the specific inoculation area (subcutaneous or intraperitoneal).

Risk of bias analysis

To gain insight into the methodological quality of the included studies, we performed a risk of bias assessment according to an adapted version of the risk of bias tool developed by Hooijmans et al. (2014b). Questions regarding reporting of randomization, blinding and sample size calculation were added to the items from the risk of bias tool (see supplemental methods for complete list). The complete list included 12 questions about the study quality such as “Was the allocation adequately concealed?” and “Were incomplete outcome data adequately addressed?” Since we were only interested in the in vivo experiments, we focused on these experiments for this assessment. Risk of bias assessment was performed by two reviewers independently (RR and WD). Differences in assessment between the reviewers were discussed until consensus was reached.

Data extraction and statistical analyses

For statistical analysis, two outcome measures that were presented frequently among the included studies were selected; survival (time-to-event data) and tumor growth inhibition.

Studies presenting survival data included experiments that show differences in survival of animals during the course of the study between the treatment conditions; chemotherapeutics administered in a DDS and chemotherapeutics administered without a DDS (free drug control). Tumor inhibition data were expressed in the studies as decrease in tumor size measured by, for instance, tumor weight or bioluminescence signal from the inoculated ovarian cancer cells.

To compare each study’s result, data was extracted from the included studies. From experiments with survival data, individual time-to-event data was extracted and from experiments with tumor growth inhibition data we extracted the raw data such as tumor weight or bioluminescence signal. If these data were only depicted graphically, authors were contacted by e-mail to provide the numerical data. If the requested data could not be provided, we extracted individual time-to-event survival data or tumor growth inhibition means with SD and the number of animals using ImageJ (1.46r, National Institutes of Health, Bethesda, MD, USA).

Since raw time-to-event survival data by themselves cannot be used for meta-analysis, hazard ratios were calculated. Hazard ratios represent the risk of dying over the course of the experiment. A hazard ratio >1 indicates that animals have a higher chance of dying due to their experimental condition, while a hazard ratio <1 indicates that animals have less chance of dying over the course of the experiment due to their treatment condition. If numerical hazard ratios were presented in included studies, they were used directly without further processing for meta-analysis. All graphically extracted survival data were first analyzed using SPSS Statistics 20.0.01 software (IBM, Amsterdam, the Netherlands). Log-hazard ratios and standard errors were determined using a Cox regression analysis with treatment conditions set as categorical covariates. Free drug control conditions (chemotherapeutic not in a DDS) were set as reference category. To compare results between studies with tumor growth outcome measures, data were translated into standardized mean differences (SMD; experimental group mean minus control group mean divided by the pooled standard deviations of the two groups). A negative SMD indicates a larger inhibition of tumor growth due to treatment with DDS compared to free drugs (not in a DDS), while a positive SMD value indicates that treatment with free drugs is more effective. Means, standard deviations (SDs) and the number of animals were extracted from the experiments and used to calculate SMDs.

Meta-analyses were performed using Review Manager Version 5.1 (Copenhagen, The Nordic Cochrane Centre, The Cochrane Collaboration, 2011). Two separate meta-analyses were performed for the outcome measures survival and tumor growth inhibition. For time-to-event data (survival), a (generic) inverse variance model with random effects and hazard ratio as effect measure was applied. In this model, the extracted log-hazard ratios with standard errors from the studies were entered in Review Manager and used to calculate hazard ratios with 95% confidence intervals for the meta-analysis. For tumor growth inhibition data, a (continuous) inverse variance model with random effects and standardized mean difference as effect measure was used. If the same study included more than two experimental conditions, the separate experiments were included in the meta-analysis. If in these cases there was only one control condition, the n for the control condition was adjusted by dividing it by the number of included conditions, to prevent that animals were included more than once in the meta-analysis. I2 was used as a measure of heterogeneity. In order to explore potential causes of heterogeneity, subgroup analyses were planned for (1) drug delivery system, (2) chemotherapeutic used, (3) xenografted cell line in animal model, (4) targeted vs. non-targeted and (5) IP vs. IV administered DDS. Because of a lack of power, subgroups containing less than three experiments were not used for subgroup analysis. To further investigate the effect of individual experiments on the overall effect or on subgroup effects, sensitivity analyses were performed by checking whether the direction of the overall or subgroup effect and their confidence intervals altered substantially when individual experiments were removed from the meta-analyses.

Furthermore, to identify possible publication bias (an underrepresentation of small studies with neutral or negative effects), a funnel scatter plot with the studies’ intervention effect on the horizontal axis and the studies’ standard error on the vertical axis was created and evaluated.

Results

Study inclusion and characteristics

Search strategies designed to include animal studies about ovarian cancer and treatment using drug delivery systems resulted in a total of 2,735 studies, whereof 1,682 and 1,053 from EMBASE and PubMed, respectively (Fig. 1). After removal of duplicates, 1,947 studies were screened by title and abstract, which resulted in removal of 1,682 studies. Subsequently, 265 studies were screened by full text. Of the studies screened by full text, 221 studies were excluded and 44 were included in this systematic review. The major reason for excluding studies was the use of a clinically irrelevant animal model (“ovarian cancer cells used in other area than peritoneal cavity or ovaries”).

Figure 1 Flow chart of study inclusion.

PubMed and EMBASE via OvidSP were searched using developed search strings to identify studies that used chemotherapeutics in a DDS in ovarian cancer animal models. All studies were first screened by title and abstract according to predefined inclusion and exclusion criteria. Subsequently studies were more specifically assessed by full text. Screenings were performed by two reviewers (RR and WD). Full text studies excluded for “others” were: (1) no full text was available or only an abstract that did not include sufficient information (n = 12); (2) conference abstract of a previously assessed full-text study (n = 5); (3) the study included only a biodistribution experiment (n = 4).

The characteristics of the included studies are summarized in Table S1. Many different DDS were designed and used to treat ovarian cancer in vivo. Most studies (36%) used liposomes to encapsulate a chemotherapeutic drug. Approximately 16% of the studies used micelles while others used capsules labeled as nano- or microparticles (9% and 27%, respectively). Furthermore, studies that used nanogels, nanosuspensions, microbullets, virus cages and nanobins were included as well. Preparation material varied among the different designed DDS as shown in Table S1. Active targeting to ovarian cancer cells using antibodies and receptor ligands such as HER-2 (Cirstoiu-Hapca et al., 2010), OV-TL3 (Storm et al., 1994; Vingerhoeds et al., 1996), folate (Chaudhury et al., 2012; Tong et al., 2014; Werner et al., 2011; Zeng et al., 2013) or luteinizing hormone-releasing hormone analogs (Pu et al., 2014) conjugated to the DDS were used in 30% of the included studies.

Several studies applied specific modifications to create a triggered drug-release. Gilmore et al. prepared nanoparticles from an acrylate monomer to create particles that are stable at neutral pH and expand after endocytosis at low pH to release their payload (Gilmore et al., 2013; Griset et al., 2009). Xu et al. (2006) prepared cisplatin nanoparticles from poly[2-(N,N-diethylamino)ethyl methacrylate]-block-poly(ethylene glycol) that also released its payload at low pH. Moreover, using a poly-isobutylene-maleic-glucosamine cisplatin combination, an acid-triggered drug delivery system was developed and probed to treat ovarian cancer by Paraskar et al. (2010) and Sengupta et al. (2012).

Other modifications were applied to ensure specific delivery and release of anti-tumor drug to ovarian cancer cells and thus to increase the efficiency of the DDS in vivo. Lu et al. (2008) designed two types of tumor penetrating microparticles from poly(DL-lactide-coglycolide) that could either prime tumors with a rapid release, or sustain a specific drug level using a slow release microparticle. Others applied a post-ultrasound strategy to release the chemotherapeutic drug from micelles or to facilitate intracellular drug uptake from microbubbles upon injection (Gao, Fain & Rapoport, 2005; Pu et al., 2014; Rapoport et al., 2004).

Frequently used clinically approved chemotherapeutic agents for ovarian cancer treatment doxorubicin, cisplatin and paclitaxel were used in 27%, 16% and 36% of the studies, respectively. The remainder used other chemotherapeutic agents as described in Table S1. One study applied co-delivery of doxorubicin and irinotecan using liposomes (Javid et al., 2014).

Other smaller parameters were applied to the DDS as well. About 32% of the included studies applied PEGylation to prolong circulation time. The route of application was varied among the included studies. DDS were either administered intraperitoneally (68%), intravenously (18%), or a combination of both (14%).

About 7% of the studies used a rat (Fisher F344, female) model in combination with the NuTu19 rat ovarian cancer cell line, while the remaining (93%) used a mouse model that was either (73%), male (2%), a combination of male and female male (2%) or not described (23%). Within the mice studies, the strains and genotypes varied a lot of which an athymic or nude (Foxn1nu) mice lacking T-cells was most frequently used (64%). Among the xenografted mice models, most were inoculated with well-established ovarian cancer cell lines OVCAR-3 (25%) or SKOV-3 (23%). Different cell numbers were inoculated in the mice, but a number of 5 ⋅ 106 cells was most frequently used. Most studies used animals that were approximately 4–8 weeks old (52%), although 41% of the studies did not describe the age of their animal model.

Risk of bias assessment

Figure 2 provides an overview of the risk of bias assessment of the 44 included studies (for scores per individual study see Supplemental Information 1). A general observation in our risk of bias assessment was that the majority of the included studies did not provide sufficient information to assess the risk of bias. The studies did not adequately describe details regarding allocation of animals to the experimental groups, adjustments for baseline differences, concealment of allocation, randomization, blinding and addressing incomplete outcome data.

Figure 2 Risk of bias analysis.

The risk of bias for all included studies was analyzed using several signaling questions. Depicted results are the answers for all studies per question.

Meta-analyses

Two types of outcome measures were frequently described in the included studies: survival and tumor growth inhibition. In order to obtain a general idea of the direction of the outcome of the different studies, meta-analyses were performed for these outcome measures separately.

Survival

Forest plots

18 studies described results with survival data. These data were used to calculate hazard ratios. A total of 30 experiments were suitable for performing a meta-analysis, which represented 377 animals. From these 30 experiments, 12 experiments showed a significantly decrease in hazard ratio, while one experiment showed a significant increase in hazard ratio (Fig. 3A). This may indicate that treatment of animal models for ovarian cancer with chemotherapeutics in a DDS is more effective in preventing death than treatment with free chemotherapeutics. For four studies (due to small group numbers) no models could be fitted, which resulted in a hazard ratio of 0 with a very wide confidence interval.

Figure 3 Effects of survival outcome measure of chemotherapeutics in a DDS compared to free chemotherapeutics (not in a DDS).

(A) The forest plot depicts hazard ratios with 95% confidence interval (CI) and the weight of the study. A hazard ratio below 1 indicates a smaller chance for the animals to die over the course of the experiment due to treatment with chemotherapeutics in a DDS. A hazard ratio higher than 1 suggests that animals have a smaller chance of dying when treated with the free chemotherapeutic control condition. Statistical significance was reached when hazard ratios with their 95% confidence interval did not include the value of 1. Numbers in brackets behind study names refer to details of the specific experiments; see Supplementary Information 1 for details. (B) Subgroup analysis for type of DDS, type of chemotherapeutic, targeted vs. non-targeted, IP vs. IV route of administration and inoculated cell type were performed. n is the number of experiments in the subgroups. I2 was used as a measure of heterogeneity.

Type of DDS

As shown in Fig. 3B subgroup analysis was performed to evaluate the overall effect of experiments that used liposomes (12 experiments) or micro/nanocapsules (15 experiments). No difference in effect on hazard ratio was found between experiments that used liposomes or micro/nanocapsules; all resulted in a significant decrease of the hazard ratio.

Type of chemotherapeutic

To investigate whether different tumor drugs encapsulated in DDS affect the hazard ratio, subgroup analysis by chemotherapeutic cisplatin (7 experiments), doxorubicin (4 experiments) and paclitaxel (16 experiments) was performed (Fig. 3B). Cisplatin, doxorubicin and paclitaxel all resulted in a significant decrease in hazard ratio. No significant differences were observed among the three drug subgroups.

Targeting vs. non-targeting

Drug delivery systems targeted specifically (12 experiments) to ovarian cancer cells did not result in a lower hazard ratio compared to non-targeted DDS (18 experiments). Both treatment strategies resulted in a lower subtotal hazard ratio, suggesting that both targeted and non-targeted DDS treatment result in improved survival rates (Fig. 3B).

Route of administration

A subgroup analysis of the different routes of administration was performed to explore whether this would affect the treatment outcome. Both IP (17 experiments) and IV (7 experiments) administration significantly lowered the risk of dying over time (Fig. 3B). Moreover, experiments that used a combination strategy of IP and IV treatment (6 experiments) also resulted in a lower hazard ratio. No statistical differences between IV, IP or a combination of IV and IP administration were observed.

Applied xenografted cell line

Ovarian cancer cell lines SKOV-3 (9 experiments), OVCAR-3 (5 experiments), A2780 (7 experiments), ID-8 (3 experiments) and IGROV-1 (3 experiments) subgroups could be included in the subgroup analysis as these had ≥3 studies in the several subgroups. This meta-analysis showed that mice xenografted with SKOV-3, OVCAR-3 and ID-8 followed by treatment with chemotherapeutics had a significant decrease in hazard ratio (Fig. 3B). Mice xenografted with IGROV-1 or A2780 that were treated with DDS did not significantly benefit from DDS treatment compared to free drug controls.

Tumor growth inhibition

Forest plot

A total of 16 studies presented data regarding tumor growth inhibition using a drug delivery system compared to a free drug control. From these studies, 21 experiments could be used for meta-analysis representing a total of 259 animals. Nine of the experiments showed a statistically significant result to the effect that chemotherapeutics in DDS inhibit tumor growth better than free drugs (Fig. 4A). The study of Konishi et al. reported a significant tumor growth inhibition. However, this could not be included in the meta-analysis due to the absence of a standard deviation in the experimental group. No studies reported significantly more tumor growth inhibition by free drug treatment compared to the DDS treatment. These results suggest that chemotherapeutics in a DDS in general have a higher efficacy regarding tumor growth inhibition than free chemotherapeutics.

Figure 4 Effects on tumor growth inhibition outcome measure of chemotherapeutics in a DDS compared to free chemotherapeutics (not in a DDS).

(A) The forest plot depicts SMDs with 95% confidence interval (CI) and the weight of the study. A statistically significant difference between interventional conditions (chemotherapeutic in DDS) and control conditions (chemotherapeutics not in a DDS) was reached when the SMD with its 95% confidence interval was greater or smaller than zero. If below zero, the interventional condition is more efficient in reducing the tumor size, while if greater than zero, the control condition is more efficient in reducing the tumor size. Numbers in brackets behind study names refer to details of the specific experiments; see Supplementary Information 1 for details. (B) Subgroup analysis for type of DDS, type of chemotherapeutic, targeted vs. non-targeted and IP vs. IV route of administration were performed. n is the number of experiments in the subgroups. I2 was used as a measure of heterogeneity.

Type of DDS

To gain insight in the effectiveness of different types of DDS, a subgroup analysis by DDS type was performed (Fig. 4B). A statistically significant difference between the subgroups micro/nano-particles (13 experiments) and micelles (3 experiments) was observed; treatment with micro/nano-particles seemed to perform better than treatment with micelles. On the other hand, no significant difference between the results of liposomes (3 experiments) and micro/nanoparticles was found.

Type of chemotherapeutic

Subgroup analysis of tumor growth inhibition data by anti-tumor drug was possible for the chemotherapeutics cisplatin (7 experiments) and paclitaxel (9 experiments) with 7 and 9 experiments, respectively (Fig. 4B). Surprisingly, cisplatin encapsulated in DDS did not result in enhanced tumor growth inhibition compared to free drug control, whereas encapsulated paclitaxel was much more effective than free paclitaxel. Moreover, the difference between subgroups paclitaxel and cisplatin was statistically significant.

Targeted vs. non-targeted

Non-targeted DDS reach tumor cells passively by exploiting the leaky vessels of the tumor vasculature. On the other hand, DDS can be decorated with tumor-specific antibodies or receptor–ligands to actively target tumor cells. A subgroup analysis for targeted (4 experiments) vs. non-targeted (17 experiments) DDS showed that both targeted and non-targeted DDS could significantly inhibit tumor growth more compared to their free drug controls (Fig. 4B). However, no significant difference was observed between the targeted and non-targeted subgroups.

Route of administration

A total of 16 experiments administered their treatment IP, while 4 experiments used an IV strategy. Both routes seem to be effective, but no statistical difference in effectiveness between the two routes was found, suggesting that IP administration of DDS has no advantage over IV in animals.

Sensitivity analysis

To assess the robustness of the meta-analyses’ results, a sensitivity analysis was performed. This analysis assessed the influence of individual studies with their specific experimental set-up (e.g., number and type of inoculated ovarian cancer cells, treatment dose and regime, or genotype differences) on the overall outcome effect.

Survival data

It was investigated whether studies that had dose differences between the DDS and free drug groups (marked with an asterisk in Figs. 3 and 4) affected the overall effect. Exclusion of these studies, however, did not affect the direction of the overall effect.

For experiments from Chaudhury et al. (2012) (one experiment), Cirstoiu-Hapca et al. (2010) (two experiments), and Yang et al. (2014) (one experiment), it was not possible to accurately estimate a hazard ratio from the log-hazard ratios. In these experiments, there was not enough information (e.g., only one event over the course of the experiment) to converge and fit a model. This resulted in a hazard ratio of 0 with a very wide confidence interval. Excluding these experiments from the analysis hardly had any effect on the overall outcome.

Tumor growth inhibition data

For tumor growth inhibition data, experiments from Javid et al. (2014) and Lu et al. (2007) showed extremely high tumor growth inhibition for their DDS groups. Therefore, we wondered whether the overall positive effect was caused by these experiments. However, these studies did only affect overall tumor growth inhibition to a small extent; a meta-analysis without these studies still resulted in a significant inhibition of tumor growth due to treatment with chemotherapeutics entrapped in a DDS.

Li & Howell (2010) and Patankar et al. (2013) used different doses of chemotherapeutics in the treatment group and control group. Therefore, it was tested whether these studies were responsible for the positive overall outcome. However, excluding these studies did not affect the overall meta-analysis effect size.

Moreover, it was investigated whether two studies that used a rat model instead of a mouse model influenced the overall outcome (Ye et al., 2013; Lu et al., 2008). A meta-analysis without these rat studies still resulted in an overall significant inhibition of tumor growth for animals treated with chemotherapeutics in a DDS compared to animals treated with free chemotherapeutics.

Publication bias assessment

Publication bias was assessed for the time-to-event outcome measure, since this outcome measure included the largest number of studies. To investigate publication bias, a funnel plot was created (Fig. 5). The experiments with almost infinite confidence intervals (Chaudhury et al., 2012; Cirstoiu-Hapca et al., 2010; Yang et al., 2014) were not included in the funnel plot as these would introduce a very large y-axis interval making the graph unclear. The funnel plot indicated missing studies at the right bottom side of the overall effect where small studies with a high hazard ratio (less survival in DDS group) would be expected, suggesting publication bias.

Figure 5 Funnel scatter plot of time-to-event studies.

Hazard ratios with a 95% confidence interval were extracted and used to create a funnel scatter plot using Review Manager. Bullets represent individual experiments from included studies. The x-axis shows the hazard ratio and the y-axis represents the standard error of the log(hazard ratio). The funnel plot is missing studies in the bottom right area in which studies with a negative outcome are expected. Since there are no studies in this area, publication bias is suggested.

Discussion

This systematic review was performed to investigate the effect of chemotherapeutic-DDS and their specific characteristics on ovarian cancer treatment in animal models. We looked at two outcome measures; survival and tumor growth inhibition, which resulted in meta-analyses of 17 and 16 studies that included 377 and 259 animals, respectively. Overall, the majority of the studies showed that treatment with chemotherapeutics entrapped in DDS used for in vivo treatment of experimental ovarian cancer had better efficacies on both survival and tumor growth inhibition compared to chemotherapeutics not entrapped in a DDS. This result is to some extent similar to what is found in clinical studies, which observed increased efficacy of doxorubicin in a DDS (pegylated liposomes) either in different staged ovarian cancer patient groups or compared to different treatment regimes with other chemotherapeutics. Although these studies did not compare free doxorubicin and doxorubicin by a DDS, most consider pegylated liposomal doxorubicin as a safe and effective treatment (Gordon et al., 2000; Muggia et al., 1997; Safra et al., 2001; Uziely et al., 1995).

However, a few observations in the field of drug delivery and ovarian cancer treatment were not supported by our results. Our results in animal studies do not show that one administration route (either IV, IP or a combination of both) had an advantage over another route looking at tumor growth inhibition and survival. This seems to be in contrast with clinical data where several lines of evidence suggest that treatment of ovarian cancer patients with a combination of IP and IV treatment with free chemotherapeutics may be more effective than IV treatment only (Jaaback, Johnson & Lawrie, 2011). It should be taken into account that these clinical studies were not performed with DDS and always included an additional systemic chemotherapy over the IP therapy. This may explain the lack of improved efficacy by IP treatment over IV treatment in our meta-analysis.

An interesting observation is that our results suggest that cisplatin, a first choice chemotherapeutic for ovarian cancer treatment, may not be a suitable candidate for treatment of ovarian cancer using DDS, since cisplatin in DDS did not lead to more tumor growth inhibition than free cisplatin. However, this was not the case for survival, a clinically more important outcome measure, where all chemotherapeutics in DDS resulted in a significant improvement of survival compared to free chemotherapeutics. It should be noted that results from tumor growth inhibition and survival outcome measures were mostly not based on data from the same studies. Interestingly, in a phase II clinical study evaluating liposomal cisplatin a lack of clinical response was observed (Seetharamu et al., 2010). Moreover, in 1998, Sugiyama et al. (1998) evaluated microspheres containing cisplatin compared to an aqueous solution of cisplatin and found in a small ovarian cancer patient group similar toxicity profiles, but no data on efficacy was shown. No subsequent phase I/II clinical trials of this DDS regarding ovarian cancer treatment could be identified in current literature, which may suggest a possible lack of clinical outcome. These two cisplatin DDS examples may confirm our results that cisplatin may not be the most suitable drug to be used in a DDS for ovarian cancer treatment.

Our results show that animal studies do not indicate higher treatment efficacies by active targeting, as both active and passive targeting resulted in almost similar inhibition of tumor growth and improved survival in animal studies. This seems to be in contrast with the current direction of the drug delivery research field where an important goal in the development of DDS is to improve treatment efficacy and simultaneously decrease side effects of chemotherapeutics. By active targeting of tumor cells with antibodies or tumor receptor ligands attached to DDS, it is hypothesized that these DDS only bind to tumor cells and not to healthy cells, thereby improving treatment efficacy and simultaneously decreasing side effects (Bae & Park, 2011). All seven included studies in our systematic review that evaluated chemotherapy by both targeted and non-targeted DDS did not show significant differences in survival or tumor reduction meta-analyses. However, if targeted therapy would show an advantage over non-targeted therapy, such as fewer side effects, chemotherapy by targeted DDS would be preferable over chemotherapy by non-targeted DDS. Nevertheless, none of the included studies showed data on reduction of side effects by targeted DDS. As our results showed no advantage of targeted DDS, although with limited power, we therefore carefully hypothesize that chemotherapy by targeted DDS may have no or only little advantage over chemotherapy by non-targeted DDS when only looking at tumor growth inhibition and survival. Future animal studies investigating differences between chemotherapy by targeted and non-targeted DDS should be performed to show the advantages of targeted DDS.

Looking at tumor growth inhibition, our analysis suggested that micro/nanoparticle DDS are most efficient and significantly better than micelles. Micelles do not result in significant tumor growth inhibition, which suggests that micelles may not be the most suitable DDS for chemotherapeutic ovarian cancer treatment. This could not be confirmed with survival data, as the micelles subgroup contained too little experiments. The two experiments evaluating micelles and showing survival data both did not show a significant improved hazard ratio. Future research should therefore show whether chemotherapy using micelles would improve survival outcome. Moreover, we would like to emphasize that the micro/nanoparticle group was very heterogeneous. However, making subgroups of the micro/nanoparticle group was not feasible due to the lack of experiments performed with each specific DDS. Therefore, more experiments containing direct comparisons would be needed to demonstrate that a specific type or class of DDS has the best efficacy.

We tried to investigate the role of the ovarian cancer animal model. During the screening of studies for inclusion in this systematic review, we came across many animal studies that used a less physiologically relevant subcutaneous animal model (Vanderhyden, Shaw & Ethier, 2003). As these animal models do not reflect the disease progression of ovarian cancer, we decided to focus only on studies that used a clinically important orthotopic intraperitoneal ovarian cancer animal model. This decision may explain why our results are less positive than the current direction in literature (e.g., no advantage of targeted DDS).

It is interesting that there is no consensus about the specific cell line used for the assessment of DDS efficacy. Domcke et al. (2013) evaluated the genetic differences between cell lines and original tumor tissue. Most frequently used ovarian cancer cells lines such as SKOV3, A2780 and IGROV-1 may not be suitable models for ovarian carcinoma cell lines and results from experiments with these cell lines should therefore be interpreted with caution, especially when translating these results to the clinic.

Our results showed no significant improved survival in animal models with A2780 or IGROV-1 cell lines. They may be considered to be poor models for ovarian cancer, but there are no explanations that these cell lines would be less sensitive for chemotherapy by DDS. Despite to their lack of clinical representativity, we have no reasons to prefer a certain cell type for experiments regarding chemotherapy by DDS based on results from this systematic review and meta-analysis.

We want to mention a number of limitations of this review. Both the overall analysis and the subgroup analyses displayed relatively high levels of heterogeneity, even though the levels within the subgroups were somewhat lower than in the overall analysis. Because of this (expected) heterogeneity, the meta-analyses were used to explore potential characteristics of DDS that affect final outcome in a hypothesis-forming rather than hypothesis-confirming manner.

Another limitation is the lack of response from authors from included studies when asked to share their raw data. As only a few authors were willing to share their raw data, we had to extract raw data from most included studies manually. Although performed carefully, this may have introduced small errors in the data used for meta-analyses.

The possibility of bias in the included studies in this systematic review may have introduced an overestimation of the meta-analyses’ results. The reliability of the results of a systematic review greatly depends on the quality of the included studies. Unfortunately, most studies lacked reporting of important details in their experimental set-up. Therefore, it was difficult to assess whether studies actually had a low or high risk of bias. To compare efficacies of chemotherapeutics in DDS compared to free chemotherapeutics, the experimental set-up is of major importance. For instance, blinding and randomization contribute to the overall validity of the experimental set-up (Hirst et al., 2014). Most studies used humane endpoints for the sake of the animals’ welfare. However, if not blinded, one can imagine that control animals may be considered to meet humane endpoint criteria earlier (Bello et al., 2014), which may introduce a bias in the outcomes of the study, particularly if survival is an outcome measure. Moreover, almost all studies used a xenografted animal model that was first inoculated with cells before treatment initiation. Without any kind of randomization, differences in tumor baseline may be introduced that could alter the final study outcome. In most of the included studies it was not mentioned that blinding or randomization was performed, which may have introduced bias (Bebarta, Luyten & Heard, 2003). Moreover, to ensure enough power of an experimental design, power calculations are an essential tool. None of the included studies described any kind of power calculation that may suggest lack of power in the included studies. These possible overestimations by studies included with bias may implicate that our observed effects may be less reliable. However, it may also be true that studies were correctly performed, but that experiments were only poorly reported, which is known from previous systematic reviews on animal studies that most studies poorly describe their in vivo experiments (Hooijmans et al., 2012). Therefore we would like to encourage to improve reporting of animals studies by using for instance the golden standard publication checklist (Hooijmans, Leenaars & Ritskens-Hoitinga, 2010) or the ARRIVE guidelines (Kilkenny et al., 2010). Finally, a funnel scatter plot analysis suggests publication bias, which could have introduced an overestimation of our results as well.

A major remark regarding our results is that we did not look at side effects as outcome measure. This aspect may change the impact of our results. For instance, IP treatment in patients results in increased survival, but these patients experience more severe side effects (e.g., pain, fatigue and gastrointestinal effects (Armstrong et al., 2006; Barlin et al., 2012; Jaaback, Johnson & Lawrie, 2011)). If the application of chemotherapeutics in DDS would decrease side effects in IP treatment, this may be a major improvement in patient quality of life. Moreover, results suggest that there is not a specific characteristic of DDS that outperforms in tumor growth inhibition or survival. Again, if a specific characteristic of DDS would show considerably less side effects, this class would be clinically very attractive although it does not outperform other DDS regarding tumor size or survival in animal studies. The same is valid for the choice of cytostatic drug. Our results do not suggest a specific higher efficacy for cisplatin, doxorubicin or paclitaxel if entrapped in a DDS regarding survival in animal studies. However, if entrapment of one of these drugs results in significant less side effects, this may be again of clinical importance and a major argument to entrap this specific chemotherapeutic in a DDS, despite similar efficacies compared to the other drugs as found in this systematic review. Although not in ovarian cancer, O’Brien showed that free doxorubicin and pegylated doxorubicin in treatment of metastatic breast cancer showed comparable overall survival with significantly less cardiotoxicity in the pegylated liposomal doxorubicin group (O’Brien et al., 2004). As only a few studies included in this systematic review addressed side-effects, an additional new systematic review on animal studies with meta-analysis should be performed to assess the specific research question; the effect of entrapment of chemotherapeutics in DDS on side effects.

In conclusion, delivery of chemotherapeutics with a DDS seems to be effective with regard to both tumor size and survival in animal models. Results of this study support the claim that delivery of chemotherapeutics is more effective compared to treatment with free chemotherapeutics, and that this efficacy is not dependent on specific characteristics of DDS. Future well-designed in vivo studies evaluating the efficacy of different characteristics of DDS on tumor size inhibition, survival and side effects should be performed to identify important characteristics of DDS for clinical translation.

Supplemental Information

Supplemental Information 1 Supplemental material

Click here for additional data file.

Supplemental Information 2 PRISMA checklist

Click here for additional data file.

Data S1 Raw data

Click here for additional data file.

Gerrie Hermkens and Jos Peeters from the Radboudumc medical library are kindly acknowledged for their help with identifying full text studies. Fang Yang and Shinju Takemoto (Dept. of Biomaterials, Radboudumc) are both greatly acknowledged for their help with the screening of Chinese and Japanese studies, respectively. Joanna in ‘t Hout (Dept. of Health Evidence, Radboudumc) is acknowledged for her statistical advice.

Additional Information and Declarations

Competing Interests

Author Contributions

Data Availability

The authors declare there are no competing interests.

René Raavé conceived and designed the experiments, performed the experiments, analyzed the data, wrote the paper, prepared figures and/or tables.

Rob B.M. de Vries conceived and designed the experiments, analyzed the data, contributed reagents/materials/analysis tools, reviewed drafts of the paper.

Leon F. Massuger reviewed drafts of the paper.

Toin H. van Kuppevelt conceived and designed the experiments and reviewed drafts of the paper.

Willeke F. Daamen conceived and designed the experiments, performed the experiments, analyzed the data, reviewed drafts of the paper.

The following information was supplied regarding data availability:

The research in this article did not generate any raw data.

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
