# Peer review of "Drug delivery systems for ovarian cancer treatment: a systematic review and meta-analysis of animal studies"

_PeerJ, doi:10.7717/peerj.1489_

## Round 0.1 · original submission · Minor Revisions

· Academic Editor

Minor Revisions

The manuscript has recieved favorable comments from the reviwers nevertheess, the reviewers have raised several comments that need to be addressed

Reviewer 1 ·

Basic reporting

The current submission by Raave et al adheres to the Peer J policies.
The English used by the authors is appropriate and makes a relatively long article easy to read and comprehend.
The background is presented in a clear and informative manner, with appropriate landmark citations.
The structure of the article conforms to the acceptable format.
The relevance of figure 5 and table 1 to the understanding of the article and clarification of data is questionable; authors should reconsider presenting them with the main manuscript. The rest of the figures are of acceptable quality and labeled appropriately.
The manuscript represents an appropriate unit of publication.

Experimental design

The manuscript is original and within the aims and scope of the journal.
The research question is relevant and clearly defined; the importance of the manuscript is presented.
The methods section is clear in presenting the statistical techniques used with their possible drawbacks.

Validity of the findings

Appropriate statistical tools were used for the available date.
The conclusions are directly derived from the results.
The authors elaborate on the strengths and weaknesses of their study allowing the reader to evaluate the significance of their conclusions.

Additional comments

In conclusion, in an era when every pharmaceutical company is trying to "arm" itself with a new targeted chemotherapy, it is important to exhaust drug delivery systems to improve the efficacy of the existing and future agents. This manuscript presents an acceptable analysis of the current literature and could be interesting to the readers.

Reviewer 2 ·

Basic reporting

The article addresses an important issue regarding the use of novel drug delivery systems for chemotherapy treatment in ovarian cancer, which is one of the most lethal gynecologic malignancies. The authors conducted a profound review of the literature in order to assess the efficacy of the use of these delivery systems compared to systemic chemotherapy.
The article adheres to PeerJ policies, and is written in a clear language with eloquent explanations.
The introduction section is clear and gives a short overview of different aspects in the studies involving the use of DDS in general and in ovarian cancer specifically, and of the role of conducting a systematic review and meta-analysis studying the data accumulated thus far on the subject.
Figures are relevant to the content of the article, and are adequately labeled.

Experimental design

The methods section is adequately divided into sub-sections. However this part of the article is too lengthy and requires some modification. For example, in the section concerning "search strategy inclusion and exclusion criteria", which gives a lengthy description of all exclusion criteria some of them obvious to the reader ("not ovarian cancer"), that should be shortened and more focused.

Validity of the findings

The results section is written in clear language and is adequately divided into sub-sections. All the figures are referred to in the text and are helpful in understanding the data provided.
A general comment regarding the results section is that there is inconsistency in the text – in several places the number of studies of a certain subgroup / characteristic is reported (for example PDF lines 291, 297), while in other places percentage of studies is reported (for example PDF lines 262-264) or both (for example PDF lines 270,301). This should be revised and reported consistently throughout the section.
The sub-section regarding "Study inclusion and characteristics" is again too lengthy and some parts of the text should be considered to be presented as a supplementary table (for example lines 300-314 regarding specific characteristics of the different animal models).
The sub-section regarding "risk of bias assessment" can be shortened as all information appears in figure 2. The important part of this section is the summary in lines 335-339 regarding the lack of information needed to adequately assess the risk of bias.


The discussion section outlines the main outcomes of the current study well, and discusses the limitations and possible contradictions with previous observations found in this study. Some of the discussion can be shortened, for example lines 511-522 which hold repetitions. The discussion about the limitations of the study due to small numbers as well as heterogeneity in different DDS, animal models etc. is very important and comprehensive.

Additional comments

None

Reviewer 3 ·

Basic reporting

-This article's objective is to assess the efficacy of DDS for the therapy of ovarian cancer. This meta-analysis focuses on DDS role in ovarian cancer therapy and includes a total of 33 animal studies (636 animals) describing survival and tumor growth inhibition. Due to heterogeneity between different types of DDS, subgroup analyses were performed enabling the researchers to draw conclusions regarding different DDS. They conclude that delivery of chemotherapeutics with a DDS is more effective in animal models with the exception of micelles and cisplatin.
This meta-analysis gives a good overlook of the subject and provides a well-founded basis or future research. The authors included all necessary items in the PRISMA checklist. The article is well structured, wording is clear, and language is coherent.

I have some minor comments:
- For context, it will be helpful to know the current status of research on human ovarian cancer (e.g. human cell lines).
-Line 204 and 205- "data was" -> data were

Experimental design

The statistical analysis is rigorous: the authors analyzed risk of bias, hazard ratio (in order to compare time to event survival data) and funnel plots (to estimate publication bias). The authors describe in great details the process of study choice and categorization, which enables the reader to follow the methodology.

Validity of the findings

The discussion is well organized. The authors explain in depth their observations and try to explain some of the more unexpected observations using current knowledge (for example why targeted DDS didn’t show advantage over non targeted DDS).

---

## Round 0.2 · accepted · Accept

· Academic Editor

Accept

The comments of the reviewers were addressed completely and the paper can be accepted for publication